# Beliefs, Attitudes, and Confidence to Deliver Electronic Cigarette Counseling among 1023 Chinese Physicians in 2018

**DOI:** 10.3390/ijerph16173175

**Published:** 2019-08-30

**Authors:** Yitian Feng, Fan Wang, Abu S. Abdullah, Xiaoyan Wang, Jing Wang, Pinpin Zheng

**Affiliations:** 1Department of Preventive Medicine and Health Education, School of Public Health, Key Lab of Public Health Safety of the Ministry of Education, Fudan University, Shanghai 200032, China; 2Department of Politics East China Normal University, Shanghai 200241, China; 3Global Health Program, Duke Kunshan University, Kunshan 215347, China; 4Duke Global Health Institute, Duke University, Durham, NC 27710, USA; 5School of Medicine, Department of General Internal Medicine, Boston University Medical Center, Boston, MA 02118, USA; 6Guanlan Networks Co. LTD., Hangzhou 310052, China

**Keywords:** electronic cigarette, physicians, counseling, knowledge, attitude, confidence

## Abstract

*Background:* The use of electronic cigarettes (e-cigarettes) is gaining popularity, so it is important to evaluate physicians’ understanding of e-cigarettes. This study assessed the beliefs, attitudes, and confidence in e-cigarette counseling among Chinese physicians and explored the factors related to asking patients about e-cigarette use. *Methods*: Physicians from across China were invited to participate in a questionnaire survey using the platform provided by DXY (www.dxy.cn) in 2018. In total, 1023 physicians completed the online survey. Descriptive analyses were used to characterize the participants, and multivariate logistic regression analyses were applied to identify predictors of physicians’ asking about patients’ e-cigarette use. *Results*: Only 46.3% of respondents agreed that e-cigarettes had adverse health effects, and 66.8% indicated that e-cigarettes can be regarded as a type of smoking cessation treatment. We found that 61.3% thought it was important to discuss e-cigarettes with patients, and 71.7% reported feeling confident about their ability in counseling about e-cigarettes. Respondents who had used e-cigarettes (OR = 2.05; 95% CI: 1.16–2.63), had received training about e-cigarettes (OR = 3.13; 95% CI: 2.17–4.52), or were confident about their ability to answer patients’ question about e-cigarettes (OR = 2.45; 95% CI: 1.65–3.65) were more likely to ask patients about e-cigarette use. Physicians who showed a supportive attitude toward using e-cigarettes to quit smoking (OR = 0.79; 95% CI: 0.63–0.99) were less likely to ask about patients’ e-cigarettes use frequently. *Conclusions*: Chinese physicians appeared to ignore the adverse health effects of e-cigarettes, and considered e-cigarettes as a smoking cessation treatment. Comprehensive training and regulations are needed to help physicians incorporate the screening of e-cigarette use into routine practice and provide patients truthful information as new data emerge.

## 1. Introduction

Electronic cigarettes (e-cigarettes) have changed the pattern of tobacco use worldwide after they were first introduced into the market in 2004 [1,2,3,4]. E-cigarettes are now widely available in various retail outlets and online, underscoring the growing demand. American adults’ use of e-cigarettes increased from 1.8% in 2010 to 13.5% in 2013 [5]. The International Tobacco Control (ITC) Surveys (between 2009 and 2013) showed the prevalence of e-cigarette use ranged from 10% in the UK to 12% in South Korea and 20% in Australia [6]. According to the 2015 China Adult Tobacco Survey, 3.1% of Chinese adults had tried e-cigarettes and 0.5% were actively using them [7]. In recent years, e-cigarette use is becoming increasingly popular in China [8,9], and domestic sales have also surged [10].

Whether e-cigarettes are a safe and viable alternative to conventional cigarettes or can renormalize cigarette use [11] is highly controversial. In some studies, e-cigarettes helped smokers overcome cravings and withdrawal symptoms [12]; some studies reported that e-cigarettes were less harmful than cigarettes [13,14,15]. However, many public health experts and opponents believe that many e-cigarettes still pose health risks for smokers and nonsmokers [16,17,18,19] due to the nicotine, carcinogens, and heavy metals released with the exhaled vapor from e-cigarettes. Adolescents’ use of e-cigarettes has increased dramatically [20,21], and the act of using e-cigarettes can encourage them to use conventional cigarettes or even lead to the dual use of cigarettes and e-cigarettes [22,23].

Advice to quit smoking and counseling provided by physicians improve smoking cessation attempts of smokers, and these approaches are highly recommended by the World Health Organization (WHO) [24,25,26]. The increased use of e-cigarettes has generated discussions regarding the effectiveness of these products as a smoking cessation intervention [27,28]. Physicians are encountering increasing numbers of smokers who have experience with e-cigarettes [29,30,31]. In an Australian study, 39% of pharmacy staff were asked about e-cigarettes by patients; 75% of pharmacy staff thought that patients would be interested in using e-cigarettes to quit [32]. Additionally, in the US, 92% of healthcare providers were aware of e-cigarettes, and 11% had treated adolescent patients who had used them [31]. In earlier studies, pediatric specialties, obstetricians, and gynecologists in the US lacked sufficient knowledge about e-cigarettes and expressed interest in learning more [31,33]. However, little is known about how Chinese healthcare providers perceive e-cigarettes and whether they actively communicate with their patients about e-cigarettes.

Hence, we sought to describe how Chinese physicians’ beliefs, attitudes, and confidence regarding e-cigarettes has become an emerging issue in China. Our study further examined the factors related to the frequency of doctor–patient communication on e-cigarette use.

## 2. Materials and Methods

### Study Design and Subjects

The present study was a national survey that explored the view of Chinese physicians regarding e-cigarettes. This online survey was conducted on the platform of DXY.cn, the largest online community for physicians and healthcare professionals in China with more than 5.5 million registered life science members, including 2 million physicians [34]. An online digital questionnaire was designed and launched, and it was available from 26 April to 21 May 2018.

The sample size was calculated by estimating the proportion of physicians who communicated with patients about e-cigarette use. However, no study has shown what proportion of Chinese physicians communicated with their patients. An earlier study in the US showed that 20.5% of physicians frequently communicated with patients about e-cigarettes in clinics [35]. It was estimated that 660 physicians would be needed to obtain an α level of 5% with a statistical power of 85%. Considering that only 58.4% of physicians had heard of e-cigarettes in the 2015 China Adult Tobacco Survey [7], we had to enlarge the sample size to 1130 to be accurate.

After sending electronic invitations to DXY member physicians (selected randomly by computer) via email and phone messages, a total of 1291 physicians responded and completed the online questionnaire. After excluding physicians who had never heard of e-cigarettes, the analysis included 1023 participants. The study was approved by the Ethics Committee of the School of Public Health, Fudan University (IRB#2018-04-677). All participants gave electronic informed consent.

## 3. Measures

We developed a questionnaire based on previous research with similar topics in the literature [36,37]. The questionnaire included questions regarding demographic information, physicians’ risk beliefs toward e-cigarettes, attitudes toward discussing e-cigarettes with patients, attitudes toward using e-cigarettes to quit, and confidence about e-cigarette counseling.

### 3.1. Risk Beliefs about E-Cigarettes

Risk beliefs about e-cigarettes were assessed by asking whether the respondents agreed with the following statements using the response categories of “strongly disagree” to “strongly agree”: “e-cigarettes are safer to use than conventional tobacco cigarettes,” “e-cigarettes could be a ‘gateway’ to other tobacco use,” “e-cigarettes could cause dual use of e-cigarettes and traditional tobacco,” “e-cigarettes have adverse health effects,” “exposure to secondhand e-cigarette vapor is harmful,” and “e-cigarettes are highly addictive”. These statements were used in a study to assess Korean lung cancer specialist physicians’ beliefs towards e-cigarettes [36] and a national survey conducted among US physicians to reflect their beliefs about e-cigarettes [37]. As there are many debates on whether e-cigarettes are safer to use than conventional tobacco cigarettes, we excluded this item when calculating the average score of risk beliefs about e-cigarettes.

A Likert scoring system (1 for “strongly disagree” through 4 for “strongly agree”) was used to calculate the total risk beliefs score. The average score of these five questions was estimated to show physicians’ risk beliefs about e-cigarettes.

### 3.2. Attitudes toward Discussing E-Cigarettes with Patients

Attitudes toward discussing e-cigarettes with patients were assessed by asking whether they believe it is important to discuss e-cigarettes with their patients [31]. Responses were rated on a 4-point Likert scale, ranging from “strongly disagree” to “strongly agree”.

### 3.3. Attitudes toward Using E-Cigarettes to Quit

To investigate physicians’ attitudes toward using e-cigarettes to quit, respondents were provided with the following questions (with response categories of “strongly disagree” to “strongly agree”): “e-cigarettes can decrease the number of cigarettes smoked,” “e-cigarettes can lower the risk of tobacco-related disease,” “e-cigarettes can be regarded as a type of smoking cessation treatment,” and “it is better to recommend e-cigarettes to smokers who failed to quit with conventional smoking cessation treatment” [36,37].

A Likert scoring system (1 for “strongly disagree” through 4 for “strongly agree”) was used to calculate the total attitude score. The average score of these four questions was calculated to show physicians’ attitudes toward using e-cigarettes to quit.

### 3.4. Confidence and Action about E-Cigarette Use Counseling

Participants were assessed on their confidence level about their ability to answer patients’ questions about e-cigarettes. A 4-point Likert scale was used (1—not at all confident, 2—somewhat not confident, 3—somewhat confident, and 4—very confident).

A supplementary object was used to investigate how often physicians asked their patients about e-cigarette use, and the response categories were: Never (0% of the time), rarely (1–24% of the time), sometimes (25–74% of the time), frequently (75–99% of the time), and always (100% of the time). For the analyses, we categorized the responses into low frequency (responses of 0–24% of the time) and high frequency (responses of 25–100%).

### 3.5. Demographic Information

Demographic characteristics for subjects included age, gender, specialty, physician type (resident physician, attending physician, associate chief physician, or chief physician), hospital level, region, smoking status (nonsmokers and former smokers or current smokers), e-cigarette use, and whether they had received any training about e-cigarettes (the content of the training should have included the health risks of e-cigarettes and the newest evidence about the safety and efficacy of using e-cigarettes to quit). We classified specialties into two categories: The first type comprised pneumology, cardiovascular health, and oncology because these departments generally had more patients with a smoking-related disease, and the second type consisted of the other specialties.

### 3.6. Statistical Analysis

Descriptive statistics were computed for all questions. The beliefs, attitudes, and confidence responses by smoking status of the physicians (nonsmokers and former smokers vs. current smokers) were compared using the chi-square test. Multivariate logistic regression analyses were used to explore the relationship between the frequency of asking patients about e-cigarette use and the characteristics of the participants. All the data were analyzed using IBM SPSS Statistics (IBM Corp., Armonk, NY, USA) for Windows Version 23.0. Significance was established as *p* < 0.05 (two-tailed), with a confidence limit at 95%.

## 4. Results

### 4.1. Demographic and Other Characteristics

Table 1 shows that, of the respondents, 67.6% were men, 47.9% were aged between 30 and 40 years old, and 18.6% specialized in pneumology, cardiovascular health, and oncology, which address many tobacco-related diseases. More than 50% of respondents were from tertiary hospitals, and 26.7% were chief or associate chief physicians. The respondents were from 31 of China’s 34 provinces, and 57.3% were from the eastern part of China. The proportion of current smokers was 12.4%, and 8% of physicians tried e-cigarettes before. Additionally, 84.5% of the respondents had not received any training on e-cigarettes, and 74.9% reported that they had never or rarely asked their patients about e-cigarette use.

### 4.2. Risk Beliefs, Attitudes, and Confidence of E-Cigarettes

The majority (69.6%) of physicians believed that e-cigarettes were safer to use than conventional tobacco cigarettes. Less than half of the participants thought that e-cigarettes could be “gateway” to other tobacco use. Only 49.6% of physicians believed that exposure to secondhand e-cigarette vapor was harmful, and 45.6% agreed that e-cigarettes were highly addictive. Plus, current smokers were less likely to believe that it was important to discuss e-cigarettes with patients compared to nonsmokers and former smokers. Table 2 shows the data from the survey.

Of all respondents, 68.6% believed that e-cigarettes could decrease the number of cigarettes smoked, and more than 70% thought that e-cigarettes could lower the risk of tobacco-related diseases. Most physicians (66.8%) agreed that e-cigarettes could be regarded as a type of smoking cessation treatment, and 72% stated it was better to recommend e-cigarettes to smokers who failed to quit with conventional smoking cessation aids. The majority (71.7%) reportedly were confident in their ability to answer patients’ questions about e-cigarettes.

### 4.3. Factors Associated with the Frequency of Chinese Physicians Asking Their Patients about E-Cigarette Use

The multivariate analysis (Table 3) shows that a high frequency of asking patients about e-cigarette use is significantly associated with physicians having used e-cigarettes (OR = 2.05), having received training about e-cigarettes (OR = 3.13), and being confident about their ability to answer patients’ questions about e-cigarettes (OR = 2.45). Additionally, showing a more positive attitude toward using e-cigarettes to quit (OR = 0.79) significantly decreases the odds of high frequency of asking patients about e-cigarette use. However, gender, age, specialty, smoking status, and risk beliefs of e-cigarettes are not associated with the frequency of Chinese physicians asking their patients about e-cigarette use.

## 5. Discussion

To the best of our knowledge, this is the first study to explore the risk beliefs, attitudes, and confidence about e-cigarette counseling among Chinese physicians. The main findings of this study are the following: (1) Most Chinese physicians thought it was important to discuss e-cigarettes with their patients and felt confident about their ability to answer patients’ questions about e-cigarettes; (2) a substantial proportion of physicians were not aware of the health risks of e-cigarettes and were supportive of using e-cigarettes to quit smoking; and (3) e-cigarette use, training about e-cigarettes, attitudes toward using e-cigarettes to quit, and the confidence regarding their ability to have discussions with patients about e-cigarettes influenced the frequency that physicians asked patients about e-cigarette use.

More than two-thirds of physicians (61.3%) in our study thought it was important to discuss e-cigarettes with their patients, which indicates that the majority of Chinese physicians were aware of e-cigarettes and saw the issue of e-cigarettes as a priority. This result is in accordance with other studies of healthcare professionals. For example, Dong et al. reported that 67.6% of 185 lung cancer specialists thought it was important to communicate with patients about e-cigarettes [36]. In a survey of 561 Minnesota health providers who treat adolescents, Jessica et al. reported that 69% of respondents agreed that it was important to discuss e-cigarettes with adolescent patients [31]. Moreover, most of our participants (71.6%) were confident in their ability to answer their questions about e-cigarettes, with the prior study showing that nearly 50% of US primary care providers were confident in their knowledge about e-cigarettes and ability to answer questions about them [37].

Despite high levels of confidence regarding answering patients’ questions on e-cigarettes, the results demonstrate a deficient knowledge of e-cigarettes’ health risks and a supportive attitude in the efficacy of e-cigarettes as smoking cessation aids among Chinese physicians. In our study, only 41.2% physicians agreed that e-cigarettes could induce other tobacco use, and 46.3% were aware of their adverse health effects, with 66.8% supporting the idea that e-cigarettes can be regarded as a type of smoking cessation treatment. Recent studies have shown that most of the Korean lung cancer physicians (83.8%) believe that e-cigarettes could be a gateway to other tobacco use [36], and 62.3% of US physicians are concerned that e-cigarettes have adverse health effects [37]. In addition, other studies have reported lower support for using e-cigarettes to quit smoking among physicians [36,38,39,40]. For example, only 18% of the specialties from Saint Louis University Hospital (USA) would advise e-cigarettes as a cessation aid (i.e., as nicotine-replacement therapy) to patients [39]. Samuel et al. reported that more than a third (35%) of family physicians in Kansas were not sure about the effectiveness of e-cigarettes, and 12% thought e-cigarettes were either very effective or effective smoking cessation products [38]. Furthermore, authorities and major health organizations do not recommend the use of e-cigarettes as a regular smoking cessation tool. The US Preventive Service Task Force (USPSTF) noted that the evidence is insufficient to recommend e-cigarettes for quitting, and clinicians should direct patients to other cessation aids that are proven to be effective and safe [41]. This suggestion is shared by the American Heart Association [42], the American Academy of Family Physicians [43], and The Spanish Society for Pulmonary Medicine and Thoracic Surgery [44].

The results of the present study indicate that Chinese physicians should learn more about e-cigarettes and participate in training that includes the basic information about e-cigarettes. The majority of physicians (84.5%) in our study had not received training about e-cigarettes, and we found that physicians who had received training about e-cigarettes (OR = 3.16) were more likely to ask their patients about e-cigarette use in a clinical setting. Therefore, comprehensive training should be provided for physicians, and it should include the health risks and the current evidence about the effectiveness of e-cigarettes as a smoking cessation treatment. Such training should also focus on screening patients’ e-cigarette use, especially by adolescents in clinical settings. Physicians must stay informed to provide patients truthful information as new data emerge.

The awareness about e-cigarette use and high levels of confidence regarding e-cigarette counseling, combined with the deficient knowledge of e-cigarettes’ negative health consequences among Chinese physicians, adds urgency to the need for legislative and regulatory actions that can hopefully curb nicotine exposure from e-cigarettes, particularly for our Chinese youth. Recently, WHO claimed in the Report on the Global Tobacco Epidemic, 2019 that e-cigarettes are unlikely to be harmless, and should not be promoted as a cessation aid until adequate evidence is compiled [45]. In addition, WHO advises parties of the FCTC (Framework Convention on Tobacco Control) that any form of e-cigarette advertising, promotion, and sponsorship must be regulated by the appropriate governmental body [45]. In mainland China, there is no precise regulation about e-cigarettes’ manufacture, sale, advertisement, or application in clinical practice [46], and many e-cigarette manufacturers advertise their products as harmless, safe, and effective smoking cessation aids to help smokers quit [47]. Therefore, the Chinese government should consider introducing policies on e-cigarettes, and explicitly proclaim the health risks of e-cigarettes. The government should also advertise the fact that there is not sufficient evidence for the effectiveness of e-cigarettes in quitting, in order to regulate the false propaganda of e-cigarettes in advertisement. China’s Clinical Smoking Cessation Guidelines should add content that physicians should send the clear message that e-cigarettes are not harmless, incorporate screening and counseling about e-cigarettes into routine clinical assessments, and guide patients to other evidence-based tobacco cessation approaches when they want to achieve smoking cessation with e-cigarettes.

## 6. Study Strengths and Limitations

This study has several strengths. To our knowledge, this is the first study to describe the beliefs, attitudes, and confidence to deliver e-cigarette counseling among Chinese physicians. In addition, this study provides timely insight into what factors are associated with Chinese physicians’ asking patients about e-cigarette use, which is helpful to guide physicians to screen patients for e-cigarette use.

There are at least two limitations to our study. First, the study only included physicians sampled through a website survey and so generalizing our findings to a broader base of Chinese physicians requires caution. Second, our analysis excluded physicians who had not heard of e-cigarettes, so the result may overestimate the e-cigarette usage rate, as well as the knowledge and confidence among Chinese physicians.

## 7. Conclusions

Overall, this study showed that most of the Chinese physicians surveyed thought it was important to discuss e-cigarettes with their patients, had a significant knowledge deficiency about e-cigarettes’ health risks, and were supportive about using e-cigarettes to quit smoking. The attitude about e-cigarettes and confidence to answer patients’ questions related to e-cigarettes could influence physicians to ask patients about e-cigarette use. To provide factual information to smokers and engage in e-cigarette screening, Chinese physicians need comprehensive training about the adverse health effects of e-cigarettes and how to communicate when conveying the health implications of e-cigarettes and screening patients’ e-cigarette use. Physicians should receive training in order to be informed as new evidence and new products emerge. There is potential for e-cigarette regulation and clinical practice guidelines in the future.

## Figures and Tables

**Table 1 ijerph-16-03175-t001:** Demographics and other characteristics of respondents.

Characteristics	No. (%)
**Gender**	
Female	331(32.4)
Male	692(67.6)
**Age**	
20–29	226(22.1)
30–39	490(47.9)
≥40	307(30.0)
**Specialty**	
Pneumology/Cardiovascular/Oncology	190(18.6)
Others	833(81.4)
**Physician type**	
Resident Physician	353(34.5)
Attending Physician	397(38.8)
Chief or Associate Chief Physician	273(26.7)
**Hospital level**	
Primary	120(11.7)
Secondary	275(26.9)
Tertiary	628(61.4)
**Region**	
East	586(57.3)
Middle	277(27.1)
West	160(15.6)
**Smoking status**	
Non-smoker or former smoker	896(87.6)
Current smoker	127(12.4)
**Have used e-cigarettes**	
No	941(92.0)
Yes	82(8.0)
**Received training about e-cigarettes**	
No	864(84.5)
Yes	159(15.5)
**How often have you asked patients about e-cigarettes use?**	
Never	349(34.1)
Rarely	398(38.9)
Sometimes	218(21.3)
Frequently	54(5.3)
Always	4(0.4)

**Table 2 ijerph-16-03175-t002:** Risk beliefs, attitudes, and confidence to deliver e-cigarette counseling among 1023 Chinese physicians.

Statements	All No. (%)	Nonsmokers and Former Smokers No. (%) (*n* = 290)	Current Smokers No. (%) (*n* = 733)	*p*-Values
**Risk beliefs of e-cigarettes** (“strongly agree” or “agree”)				
E-cigarettes could be a “gateway” to other tobacco use.	421(41.2)	367(41.0)	54(42.5)	0.773
E-cigarettes could cause dual use of e-cigarettes and traditional tobacco.	619(60.5)	533(59.3)	86(67.7)	0.081
E-cigarettes have adverse health effects.	474(46.3)	412(46.0)	62(48.2)	0.569
Exposure to secondhand e-cigarette vapor is harmful.	507(49.6)	448(50.0)	59(56.5)	0.507
E-cigarettes are highly addictive	467(45.6)	412(46.0)	55(43.3)	0.634
*Average score*	2.48 ± 0.67	2.48 ± 0.67	2.47 ± 0.65	0.797
**Attitudes toward discussing e-cigarettes with patients** (“strongly agree” or “agree”)				
It is important to discuss e-cigarettes with patients.	627(61.3)	564(62.9)	63(49.6)	0.005
**Attitudes toward using e-cigarettes to quit** (“strongly agree” or “agree”)				
E-cigarettes can decrease the number of cigarettes smoked.	702(68.6)	622(69.4)	80(63.0)	0.153
E-cigarettes can lower the risk of tobacco-related disease.	731(71.5)	642(71.7)	89(70.1)	0.753
E-cigarettes can be regarded as a type of smoking cessation treatments.	643(66.8)	599(66.9)	84(66.1)	0.920
It is better to recommend e-cigarettes to smokers who failed to quit with conventional smoking cessation treatment.	737(72.0)	650(72.5)	87(68.5)	0.343
*Average score*	2.86 ± 0.78	2.87 ± 0.77	2.82 ± 0.81	0.562
**Confidence** (“strongly agree” or “agree”)				
I am confident about my ability to answer patients’ questions about e-cigarettes.	733(71.7)	637(71.1)	96(75.6)	0.344

Footnote: The average score ranged from 1 to 4.

**Table 3 ijerph-16-03175-t003:** Odds of the frequency of Chinese physicians asking their patients about e-cigarette use.

Variables	Low Frequency No. (%) (*n* = 766)	High Frequency No. (%) (*n* = 257)	OR (95% CI)
**Gender**			
Female (reference)	511(66.7)	181(70.4)	1
Male	255(33.3)	76(29.6)	1.00(0.72,1.40)
**Age**			
20–29 (reference)	178(23.2)	48(18.7)	1
30–39	369(48.2)	121(47.1)	1.01(0.69,1.54)
≥40	219(28.6)	88(34.2)	1.38(0.89,2.13)
**Hospital level**			
Primary (reference)	91(11.9)	29(11.3)	1
Secondary	201(26.2)	74(28.8)	1.04(0.62,1.78)
Tertiary	474(61.9)	154(59.9)	0.92(0.56,1.48)
**Physician type**			
Resident Physician (reference)	273(35.6)	80(31.1)	1
Attending Physician	293(38.3)	104(40.5)	1.04(0.61,1.75)
Chief or Associate Chief Physician	200(26.1)	73(28.4)	0.92(0.57,1.27)
**Region**			
East (reference)	439(57.3)	147(57.2)	1
Middle	206(26.9)	71(27.6)	1.01(0.71,1.42)
West	121(15.8)	39(15.2)	0.88(0.58,1.36)
**Specialty**			
Pneumology/Cardiovascular/Oncology (reference)	139(18.1)	51(19.8)	1
Others	627(81.9)	206(80.2)	0.87(0.60,1.27)
**Smoking status**			
Nonsmokers or former smoker (reference)	670(87.5)	226(87.9)	1
Current smoker	96(12.5)	31(12.1)	0.65(0.38,1.10)
**Have used e-cigarettes ***			
No (reference)	716(93.5)	225(87.5)	1
Yes	50(6.5)	32(12.5)	2.05(1.16,2.63)
**Have received training about e-cigarettes’ health risks ****			
No (reference)	684(89.3)	180(70.0)	1
Yes	82(10.7)	77(30.0)	3.13(2.17,4.52)
**Risk beliefs of e-cigarettes**	3.048(0.569)	3.052(0.598)	0.80(0.62,1.03)
**It is important to discuss e-cigarettes with the patients**			
No (reference)	313(40.9)	83(32.3)	1
Yes	453(59.1)	174(67.7)	1.37(0.98,1.91)
**Attitudes toward using e-cigarettes to quit ***	2.852(0.770)	2.883(0.791)	0.79(0.63,0.99)
**Confident about my ability to answer patients’ questions about e-cigarettes ****			
No (reference)	251(32.8)	39(15.2)	1
Yes	515(67.2)	218(84.8)	2.45(1.65,3.65)

Footnote: CI is the Confidence Interval; OR is the Odds Ratio. * *p* < 0.05, ** *p* < 0.01.

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
