# Peer review of "Beliefs, Attitudes, and Confidence to Deliver Electronic Cigarette Counseling among 1023 Chinese Physicians in 2018"

_ijerph, 2019, doi:10.3390/ijerph16173175_

Round 1

Reviewer 1 Report

 Knowledge, Attitudes, and Confidence to Deliver 2 Electronic Cigarettes

Counseling Among 1023 3 Physicians, China, 2018

Given the debate on e-cigarette, it might be useful for the authors to make a conclusion  or a position on e-cigarettes to guide the whole paper and the discussion. It seems to go back and forth and no clear direction on whether they think e-cigarettes are safe and efficacious for cessation or not.

Need to be distinctive about the group being referred to rather than providing general sentences. For Example “talking to patients about e-cigarettes” is it all patients or those who use tobacco?

Abstract

Page 1 lines 16-17, could clarify what the authors mean by “gaining popularity” for example is it popularity about awareness, use, disagreements etc.

 With electronic cigarettes (e-cigarettes) gaining popularity recently, it is important to find out what is currently being discussed about e-cigarettes between Chinese physicians and patients.

Page 1 lines 18-20 – could clarify what “screening on e-cigarettes use” mean.

This study assessed the knowledge, attitudes and confidence of e-cigarettes 18 counseling among Chinese physicians and explored the factors related to their screening on e-cigarettes use.

Page 1 lines 28-29 – could clarify what is meant by “training about e-cigarettes”… or details of training.

 Respondents who had received training about e-cigarettes (OR=3.16; 95% CI: 2.19-28 2.76), held more important attitude in communicating about e-cigarettes

Page 1 lines 29 – what is meant by “held more important attitude in communicating about e-cigarettes” what are considered as more important in this line.

Page 1 line 30 – what is meant by “e-cigarettes counseling”

 were confident about their ability to e-cigarettes counseling

Page 1 lines 34-36 – there is still on-going debate on safety of e-cigarettes and as cessation tools – how could this be feasible to ensure Physicians learn about the evidence-based counseling about e-cigarettes.

 Physicians need to learn more about the safety of e-cigarettes use and efficacy of e-cigarettes in quitting to ensure that patients receive evidence-based counseling about e-cigarettes.

Introduction

Page 1 lines 40-41 – the sentence makes reference to “worldwide” but the references cited only refer to the UK.

 Electronic cigarettes (e-cigarettes) are changing the pattern of tobacco use worldwide since they 40 were first introduced into the market in 2004[1,2].

Page 2 line 47 – reference 6 does not show that e-cigarette use is becoming increasingly popular in China. It explains more of the marketing tactics but does not provide estimates to show the increase in use of e-cigarettes in China. Consider other references to support you sentence.

In recent years, e-cigarette use is becoming increasingly popular in China[6],

Page 2 line 60-61 - suggest to provide reference/s for this sentence

The increased use of e-cigarettes has generated discussion regarding the effectiveness of these 60 products as a smoking cessation intervention.

Page 2 lines 66-67 – suggest you provide specific reference group or country here -

In earlier studies, healthcare 66 providers lacked sufficient knowledge about e-cigarettes and express interest in learning more[28,29].

Materials and Method

Page 2 lines 80-81 – suggest to explain how was the proportion of physicians who communicate with patients was estimated?

The sample size was calculated by estimating the proportion of physicians who communicate with patients about e-cigarettes use.

Page 2 lines 86-87 – calculated sample was 660 and was bumped up to 800 to “accurate”. Why is that the study recruited 1291 physicians which is more than the sample size you planned for. Secondly, did all the physicians contact receive both email and phone message?

After sending the electronic invitations to DXY member physicians randomly via email and 86 phone message, we recruited 1291 physicians to complete the online questionnaire.

Page 3 lines 95 – clarify “physicians’ knowledge toward e-cigarettes” and “attitudes towards e-cigarettes communication”. Both are not clear.

physicians’ knowledge toward e-cigarettes, attitudes toward e-cigarettes communication

Page 3 lines 113 – was it all patients or only those who use tobacco?

whether they believe it is important to discuss e-cigarettes with their patients

Page 3 lines 116 – Is it using e-cigarettes as a replacement to quit or using e-cigarettes to help quit?

To investigate the physicians’ attitudes toward using e-cigarettes as a replacement to quit, respondents were provided with the following questions

Page 3 lines 132-133 – what is the justification or rationale for using this scale?

For the analyses, we categorized the responses into low frequency (responses of 0-24% of the time) and high frequency (responses of 25-100%).

Results

Page 4 lines 155-156 – what is referred to as “formal training on e-cigarettes” and do physicians have to ask all their patients about e-cigarettes.

Additionally, 84.5% of the respondents had not received any formal training on e-cigarettes, and 74.9% reported that they had never or rarely asked their patients about e-cigarettes use.

In Table 1- Please define “Received training about e-cigarettes

In Table 1 – may consider separating “never” and “rare” in the question “How often have you asked patients about e-cigarettes use?

Page 5 lines 160-161 and 168-169

Would suggest reporting the results in this section and not making conclusions and discussions as in the sentence

Our results showed that information on health hazards of e-cigarettes among Chinese physicians  was limited.

Table 2 shows that Chinese physicians possessed a supportive attitude toward using e-cigarettes to quit and felt confident in e-cigarettes counseling

Page 5 lines 163-164 – does the survey measure “knew that exposure to secondhand e-cigarettes vapor was harmful or their belief that that exposure to secondhand e-cigarettes vapor was harmful”.

Only 49.6% of physicians knew that exposure to secondhand e-cigarettes vapor was harmful, and 45.6% agreed that e-cigarettes were highly addictive.

Page 6 lines 181 – suggest provide evidence that this is a fact.

believing the fact that communicating with patients about e-cigarettes is important

Discussion

Page 7 lines 189 – paper refers to national sample – the study did not use a national sample given the exclusion criteria and in addition, the frame used to draw the sample may not be representative of the whole nation.

Page 7 lines 189-191 – the study may only in a limited way determine the training needs to help Chinese physicians addressing e-cigarettes in their practices. In addition, it is not clear what is meant by addressing e-cigarettes as it might refer to helping those who use e-cigarettes to quitting using them or advising smokers on using e-cigarette as a cessation tool. The challenge is also that there is still an on-going debate on the safety and efficacy of e-cigarettes as cessation tool.

The findings should have implications in determining the training need to help Chinese physicians addressing e-cigarettes in their practice and for developing clinical guidelines about e-cigarettes in China.

Page 7 lines 192-194 – the population in this study are practicing physicians – however, the reference group in the discussion from Germany, Hungary, Pakistan and, US are medical students.

Page 7 lines 198-199 – this is a challenging conclusion to make given that evidence is still evolving on the safety or e-cigarettes and their efficacy as cessation tools. It may be difficult to provide more education in China where the government has no clear policy on e-cigarettes. For example can physicians be educated that e-cigarettes are cessation tool when there is still debate on this?

It is evident that Chinese physicians require knowledge about the potential health hazards of e-cigarettes, indicating more education about e-cigarettes is needed.

Page 7 lines 201-202 – who are the other in the sentence below

Thereare remaining gaps between the knowledge of e-cigarettes for Chinese physician comparing with other.

Page 8 lines 209 – provide reference/s to support this assertion

may be due to their perceptions of e-cigarettes to help patients quit smoking.

Patients – in the discussion, you may want to clarify if the reference is all patients or only patients who smoke and former smokers

Page 8 lines 226-228 – the starting point is not regulating the media but have clear national policy on e-cigarettes that will provide a basis on regulating the media. If national policy indicate e-cigarettes are not safe, then the media can be regulated to report that way.

These unregulated promotional activities create misunderstandings about 226 the benefits of e-cigarettes use among the public and even among the physicians, underscoring the 227 need for strict regulatory measures of media or online advertisements.

Page 8 lines 230-231 – it is still not clear what training on e-cigarettes should cover and where supportive of their use of prevention of their use given the evolving debate on the products

Furthermore, we found that physicians who have received training about e-cigarettes

Page 8 lines 235-237 – what are these misconceptions about e-cigarettes – may provide reference/s

Such training should focus to increase physicians’ counseling competency about e-cigarettes and improve screening of its use in clinical settings, which is essential to raise the awareness 236 of adverse health effects and eliminate misconception about e-cigarettes among the public.

Strengths and Limitations

Sample is not representative of all physicians in China given the selection criteria – and limitations of the sampling frame

May need to define what is meant by “Screening on e-cigarettes”

Conclusion

Page 10  - it might be useful to clarify on knowledge definite about e-cigarettes – how is this define and similarly what is training about e-cigarettes – what are the content – or provide references to this effect

Overall, this study showed that Chinese physicians had a significant knowledge deficit about e-cigarettes, and a large proportion of them did not receive any training about e-cigarettes.

Page 10 lines 258-261 – from this conclusion, it is not clear what content to add on e-cigarettes to the Chinese smoking cessation clinical guideline – and may want to define what screening on smokers on e-cigarettes use mean and also define professional cigarettes counseling.

In addition, the new version of Chinese smoking cessation clinical guideline should add the content on e-cigarettes including the importance of screening smokers on e-cigarettes use, and how to deliver the professional e-cigarettes counseling.

Author Response

Dear reviewer:

We are thankful for your comments and suggestions. They are helpful for revising and improving our paper, as well as the important guiding significance to our researches. Based on your comments and suggestions, we have made modification on our manuscript and re-written the discussion section. We have confirmed the conclusion that Chinese physicians should take the advice from WHO, and e-cigarette should not be promoted as a cessation aid until adequate evidence is compiled. Furthermore, we have tried our best to be distinctive about the group being referred. As for language, our manuscript has been polished by LetPub during revision, which is a professional company addressing language issues of papers.

Responds to your comments:

Abstracts

(1) Page 1 lines 16-17, could clarify what the authors mean by “gaining popularity” for example is it popularity about awareness, use, disagreements etc.

Re: Thanks for your suggestion. We have modified it as you suggested.

P1,L16-17: The use of electronic cigarettes (e-cigarettes) is gaining popularity, so it is important to evaluate physicians’ understanding of e-cigarettes.

(2) Page 1 lines 18-20 – could clarify what “screening on e-cigarettes use” mean.

Re: Screening on e-cigarette use means that physicians ask about e-cigarette use in clinics, and we have clarified it in the abstract.

P1, L18-20: This study assessed the beliefs, attitudes and confidence in e-cigarette counseling among Chinese physicians and explored the factors related to asking patients about e-cigarette use.

(3) Page 1 lines 28-29 – could clarify what is meant by “training about e-cigarettes” or details of training.

Re: We are sorry about didn’t clarify the meaning before. The content of such training should include the health risks of e-cigarettes and the newest evidence about the safety and efficacy of using e-cigarettes to quit. We also have confirmed the meaning of training in the method section.

(4) Page 1 lines 29 – what is meant by “held more important attitude in communicating about e-cigarettes” what are considered as more important in this line.

Re: We are very sorry that we didn’t make the meaning clear. “Held more important attitude in communicating about e-cigarettes” means physicians thought it was important to discuss with patients about e-cigarettes. We have made some modification in the manuscript.

(5) Page 1 line 30 – what is meant by “e-cigarettes counseling”

Re: “E-cigarette counseling” means answering patients’ questions about e-cigarettes. We have modified it in the manuscript.

P1, L30: Respondents who had used e-cigarettes (OR=2.05; 95% CI: 1.16-2.63), had received training about e-cigarettes (OR=3.13; 95% CI: 2.17-4.52), were confident about their ability to answer patients’ question about e-cigarettes (OR=2.45; 95% CI: 1.65-3.65), were more likely to ask patients about e-cigarette use.

(6) Page 1 lines 34-36 – there is still on-going debate on safety of e-cigarettes and as cessation tools – how could this be feasible to ensure physicians learn about the evidence-based counseling about e-cigarettes.

Re: Thanks for your question. Recently, WHO gave advices about this issue in Report on the Global Tobacco Epidemic 2019, and Chinese physicians should be informed that e-cigarette is not harmless, and could not be be promoted as a cessation aid until evidence is sufficient. We have modified the sentence here.

Comprehensive training and regulations are needed to help physicians incorporate the screening of e-cigarette use into routine practice and provide patients truthful information as new data emerge.

Introduction

(1) Page 1 lines 40-41 – the sentence makes reference to “worldwide” but the references cited only refer to the UK.

Re: We sincerely apologize for our careless mistakes, and we have added other studies of Japan, Australia, and New Zealand.

(2) Page 2 line 47 – reference 6 does not show that e-cigarette use is becoming increasingly popular in China. It explains more of the marketing tactics but does not provide estimates to show the increase in use of e-cigarettes in China. Consider other references to support you sentence.

Re: We are sorry about this mistake, and we have changed the reference to “Li SS, Xiao D, Chu SL, Qin HY, Wang C. A survey of E-cigarette consumption in Beijing. Chinese Clinicians. 2015;43(3):47–49” and “Electronic cigarette use and smoking cessation behavior among adolescents in China.”

(3) Page 2 line 60-61 - suggest to provide reference/s for this sentence

Re: Considering your suggestion, we have added reference named “Ghosh, S.; Drummond, M.B. Electronic cigarettes as smoking cessation tool: are we there? CURR OPIN PULM MED 2017, 23, 111-116.” and “Leduc, C.; Quoix, E. Is there a role for e-cigarettes in smoking cessation? THER ADV RESPIR DIS 2016, 10, 130-135.”

P2 L60-61: The increased use of e-cigarettes has generated discussions regarding the effectiveness of these products as a smoking cessation intervention [27,28].

(4) Page 2 lines 66-67 – suggest you provide specific reference group or country here

Re: Thanks for your suggestion. We have added the reference group and country in the manuscript.

P2 L66-67: In earlier studies, pediatric specialties, obstetricians and gynecologists in the US lacked sufficient knowledge about e-cigarettes and expressed interest in learning more.

(5) Page 2 lines 80-81 – suggest to explain how was the proportion of physicians who communicate with patients was estimated?

Re: Though no study have shown that proportion among Chinese physicians, we found that 20.5% of US physicians communicated with patients about e-cigarettes frequently in the clinic in the previous study, so we used 20.5% to calculate the sample size.

(6) Page 2 lines 86-87 – calculated sample was 660 and was bumped up to 800 to “accurate”. Why is that the study recruited 1291 physicians which is more than the sample size you planned for. Secondly, did all the physicians contact receive both email and phone message?

Re: Thanks for your question. We have checked and re-written the part of calculating sample size. We have to enlarge 660 to 1130, because 2015 China Adult Tobacco Survey showed that only 58.4% of physicians had heard of e-cigarettes. Besides, all the physicians received email and phone messages.

P2 L86-87: It is estimated that 660 physicians are needed to obtain an α level of 5% with a statistical power of 85%. Considering that only 58.4% of physicians had heard of e-cigarettes in the 2015 China Adult Tobacco Survey [7], we had to enlarge the sample size to 1130 to be accurate.

(7) Page 3 lines 95 – clarify “physicians’ knowledge toward e-cigarettes” and “attitudes towards e-cigarettes communication”. Both are not clear.

Re: Based on your request, we have change them to “risk beliefs of e-cigarettes” and “attitudes toward discussing e-cigarettes with patients”.

(8) Page 3 lines 113 – was it all patients or only those who use tobacco?

Re: Thanks for your question. It is to all patients, not only for those who use tobacco.

(9) Page 3 lines 116 – Is it using e-cigarettes as a replacement to quit or using e-cigarettes to help quit?

Re: Thanks for your question. It is using e-cigarettes as a replacement to quit

(10) Page 3 lines 132-133 – what is the justification or rationale for using this scale?

Re: Thanks for your question. This scale has been used in many studies before, such as, “Beliefs, Practices, and Self-efficacy of US Physicians Regarding Smoking Cessation and Electronic Cigarettes: A National Survey” and “Lung cancer specialist physicians' attitudes towards e-cigarettes: A nationwide survey”. Therefore, we think these questions can reflect the beliefs, attitudes and confidence toward e-cigarettes among physicians.

Results

(1) Page 4 lines 155-156 – what is referred to as “formal training on e-cigarettes” and do physicians have to ask all their patients about e-cigarettes.

Re: Thanks for your question. As we mentioned before, the training of e-cigarettes means that the content of such training should include the health risks of e-cigarettes and the newest evidence about the safety and efficacy of using e-cigarettes to quit. And physicians should ask all their patients about their e-cigarette use.

(2) In Table 1- Please define “Received training about e-cigarettes

Re: Thanks for your question, and we have explained the meaning in the previous answers.

(3) In Table 1 – may consider separating “never” and “rare” in the question “How often have you asked patients about e-cigarettes use?

Re: Thanks for your suggestion. We have showed each category of this question in table 1.

How often have you asked patients about e-cigarettes use?

Never

Rarely

Sometimes

Frequently

Always

349(34.1)

398(38.9)

218(21.3)

54(5.3)

4(0.4)

(4) Page 5 lines 160-161 and 168-169

Would suggest reporting the results in this section and not making conclusions and discussions as in the sentence

Re: Thanks for your suggestion. We have deleted the sentences you mentioned.

(5) Page 5 lines 163-164 – does the survey measure “knew that exposure to secondhand e-cigarettes vapor was harmful or their belief that that exposure to secondhand e-cigarettes vapor was harmful”.

Re: Thanks for your question. We want to measure physicians’ belief about “exposure to secondhand e-cigarettes vapor is harmful”, and we have modified the sentences here.

P5 L163-164: Only 49.6% of physicians believed that exposure to secondhand e-cigarette vapor was harmful, and 45.6% agreed that e-cigarettes were highly addictive.

(6) Page 6 lines 181 – suggest provide evidence that this is a fact.

Re: We are sorry about inaccurate expression of this result, and we have modified this sentence below.

P5 L181: The multivariate analysis (Table 3) shows that a high frequency of asking patients about e-cigarette use is significantly associated with physicians having used e-cigarettes (OR=2.05), having received training about e-cigarettes (OR=3.13), and being confident about their ability to answer patients’ questions about e-cigarettes (OR=2.45).

Discussion

(1) Page 7 lines 189 – paper refers to national sample – the study did not use a national sample given the exclusion criteria and in addition, the frame used to draw the sample may not be representative of the whole nation.

Re: Thanks for your suggestion, we agree with you and have modified the beginning of discussion.

P7 L189: To the best of our knowledge, this is the first study to explore the risk beliefs, attitudes, and confidence about e-cigarette counseling among Chinese physicians.

(2) Page 7 lines 189-191 – the study may only in a limited way determine the training needs to help Chinese physicians addressing e-cigarettes in their practices. In addition, it is not clear what is meant by addressing e-cigarettes as it might refer to helping those who use e-cigarettes to quitting using them or advising smokers on using e-cigarette as a cessation tool. The challenge is also that there is still an on-going debate on the safety and efficacy of e-cigarettes as cessation tool.

Re: Thanks for your question. Although there is a debate on whether e-cigarettes are a safe and efficacious cessation tool, WHO gave advices about this issue in Report on the Global Tobacco Epidemic 2019. We found that a deficient knowledge of e-cigarettes’ health risks and a supportive attitude in the efficacy of e-cigarettes as smoking cessation aids among Chinese physicians. Therefore, comprehensive trainings are needed to provide more information about e-cigarettes for physicians. We rewrote this part of discussion here.

Therefore, comprehensive training should be provided for physicians, and it should include the health risks and the current evidence about the effectiveness of e-cigarettes as a smoking cessation treatment. Such training should also focus on screening patients’ e-cigarette use, especially by adolescents in clinical settings. Physicians must stay informed to provide patients truthful information as new data emerge.

In addition, WHO advises parties of the FCTC (Framework Convention on Tobacco Control) that any forms of e-cigarette advertising, promotion and sponsorship must be regulated by the appropriate governmental body [46]. In mainland China, there is no precise regulation about e-cigarettes’ manufacture, sale, advertisement or application in clinical practice [47], and many e-cigarette manufacturers advertise their products as harmless, safe and effective smoking cessation aids to help smokers quit [48].

(3) Page 7 lines 192-194 – the population in this study are practicing physicians – however, the reference group in the discussion from Germany, Hungary, Pakistan and, US are medical students.

Re: It is really true as you suggested that reference group are medical students, and these data couldn’t be compared with the results of our research. After discussion, we thought that the rate of e-cigarette use does not relate to our study’s focus, so we delete this paragraph.

(4) Page 7 lines 198-199 – this is a challenging conclusion to make given that evidence is still evolving on the safety or e-cigarettes and their efficacy as cessation tools. It may be difficult to provide more education in China where the government has no clear policy on e-cigarettes. For example, can physicians be educated that e-cigarettes are cessation tool when there is still debate on this?

Re: Thanks for your question. Although there are no regulations in China, the WHO's advice in Report on the Global Tobacco Epidemic 2019 is worth referring to for Chinese physicians. We hope physicians could convey the health implications of e-cigarettes and screen patients’ e-cigarette use after being informed the health risks and the current evidence about the effectiveness of e-cigarettes as a smoking cessation treatment.

(5) Page 7 lines 201-202 – who are the other in the sentence below

Re: Thanks for your question. We rewrote this paragraph and showed the results from other studies in details.

In our study, only 41.2% physicians agreed that e-cigarettes could induce other tobacco use, and 46.3% were aware of their adverse health effects, with 66.8% supporting the idea that e-cigarettes can be regarded as a type of smoking cessation treatment. Recent studies showed that most of the Korean lung cancer physicians (83.8%)believe that e-cigarettes could be a gateway to other tobacco use [37], and 62.3% of US physicians are concerned that e-cigarettes have adverse health effects[38]. In addition, other studies reported lower support for using e-cigarettes to quit smoking among physicians[37,39-41]. For example, only 18% of the specialties from Saint Louis University Hospital (USA) would advise e-cigarettes as cessation aid (i.e, as nicotine-replacement therapy) to patients [40]. Samuel et al. reported that more than a third (35%) of family physicians in Kansas were not sure about the effectiveness of e-cigarettes, and 12% thought e-cigarettes were either very effective or effective smoking cessation products [39]. (7) Page 8 lines 226-228 – the starting point is not regulating the media but have clear national policy on e-cigarettes that will provide a basis on regulating the media. If national policy indicate e-cigarettes are not safe, then the media can be regulated to report that way.

(6) Page 8 lines 209 – provide reference/s to support this assertion.

Re: We are sorry that could not provide reference to support this opinion, so we delete this sentence.

(7) Page 8 lines 226-228 – the starting point is not regulating the media but have clear national policy on e-cigarettes that will provide a basis on regulating the media. If national policy indicate e-cigarettes are not safe, then the media can be regulated to report that way.

Re: Thanks for your question, we are definitely agree with your idea. So, we have modified the sentences here.

In mainland China, there is no precise regulation about e-cigarettes’ manufacture, sale, advertisement or application in clinical practice [47], and many e-cigarette manufacturers advertise their products as harmless, safe and effective smoking cessation aids to help smokers quit [48]. Therefore, the Chinese government should consider introducing policies on e-cigarettes, and explicitly proclaim the health risks of e-cigarettes, the government should also advertise the fact that there is not sufficient evidence for the effectiveness of e-cigarettes in quitting, in order to regulate the false propaganda of e-cigarettes in advertisement.

(8) Page 8 lines 230-231 – it is still not clear what training on e-cigarettes should cover and where supportive of their use of prevention of their use given the evolving debate on the products.

Re: Thanks for your question, and we have explained the meaning in the previous answers. In addition, we have re-written these sentences.

Such training should also focus to increase physicians’ communication competency in conveying the complexity and current uncertainty about the health implications of e-cigarettes, and improve screening of its use, especially by adolescent patients in clinical settings. Physicians must stay informed to provide patients truthful information as new data emerge. What’s more, China Clinical Smoking Cessation Guidelines should add the content that physicians should send the clear message that e-cigarettes are not harmless, guide patients to other evidence-based tobacco cessation approaches when they want to achieve smoking cessation with e-cigarettes.

(9) Page 8 lines 235-237 – what are these misconceptions about e-cigarettes – may provide references

Re: Thanks for your question. The misconception is that e-cigarette is harmless and is safe as a smoking cessation tool. We had modified this paragraph and delete “misconception”.

The government should also advertise the fact that there is not sufficient evidence for the effectiveness of e-cigarettes in quitting, in order to regulate the false propaganda of e-cigarettes in advertisement. China’s Clinical Smoking Cessation Guidelines should add content that physicians should send the clear message that e-cigarettes are not harmless, incorporate screening and counseling about e-cigarettes into routine clinical assessments, and guide patients to other evidence-based tobacco cessation approaches when they want to achieve smoking cessation with e-cigarettes.

Strengths and Limitations

Sample is not representative of all physicians in China given the selection criteria – and limitations of the sampling frame

Re: Based on your suggestion, we have made some modification about the strengths section.

To our knowledge, this is the first study to describe the beliefs, attitudes, and confidence to deliver e-cigarette counseling among Chinese physicians.

Conclusion

(1) Page 10- it might be useful to clarify on knowledge definite about e-cigarettes – how is this define and similarly what is training about e-cigarettes – what are the content – or provide references to this effect.

Re: Thanks for your suggestion, we have clarified the meaning you mentioned.

P10: To provide factual information to smokers and engage in e-cigarette screening, Chinese physicians need comprehensive training about the adverse health effects of e-cigarettes and how to communicate when conveying the health implications of e-cigarettes and screening patients’ e-cigarette use. Physicians should receive training in order to be informed as new evidence and new products emerge. There is potential for e-cigarette regulation and practice guidelines in the future.

(2) Page 10 lines 258-261 – from this conclusion, it is not clear what content to add on e-cigarettes to the Chinese smoking cessation clinical guideline – and may want to define what screening on smokers on e-cigarettes use mean and also define professional cigarettes counseling.

Re: Thanks for your suggestion, we agree with you and have modified this section.

P10 L258-261: To provide factual information to smokers and engage in e-cigarette screening, Chinese physicians need comprehensive training about the adverse health effects of e-cigarettes and how to communicate when conveying the health implications of e-cigarettes and screening patients’ e-cigarette use. Physicians should receive training in order to be informed as new evidence and new products emerge. There is potential for e-cigarette regulation and clinical practice guidelines in the future.

Reviewer 2 Report

This is an important study that investigated the views that physicians in China have about e-cigarettes. I have a few suggestions for minor/major edits below:

Introduction…this section did a good job of introducing the topic and the need for the research study

Methods…line 88…change “is” to “was” and “answer” to “answered” for correct tense. Also, please edit the end of the sentence: “…as end this study” it is hard to understand.

Methods…lines 108-110 (and Table 2)…since Likert items are ordinal measures, they cannot be statistically “averaged.” Please remove the average scores from your analysis and table. The findings from each item in the table are interesting enough, without needing to provide an average that is not possible statistically.

Methods…I recommend deleting the analysis that includes never vs. ever smoking status. This doesn’t relate to your research study’s focus. Also, studies that ask smoking status of healthcare providers tend to look at current smoking status, not “ever” smoking. Ever smoking includes just trying a cigarette one time, which is much different that if the healthcare provider was a current smoker.

Results…Table 2…please show the frequencies and percentages of agree, neutral, disagree for each Likert item.

Discussion…lines 208-228…this entire paragraph should be deleted. This paragraph is based on your assumption that physicians think it’s important to discuss e-cigarettes with patients as a cessation tool. However, the survey item did not go into that level of detail. It simply asked about discussing e-cigarettes. The physicians could have interpreted the survey item as being important to discuss e-cigarettes because they are harmful.

Discussion…discussion sections should discuss which findings from the study are worthy of discussion. So far, you have discussed that 8% of physicians used e-cigarettes, a paragraph based on an assumption that should be deleted, and a final paragraph that calls for more education for physicians about e-cigarettes. Please add a few paragraphs regarding the knowledge and attitudes that are concerning. These are important findings that should be discussed.

Discussion…limitations…lines 248-249…the limitation is not about cause and effect, the limitation is that you had a convenience sample of physicians sampled through a website. Because it is a convenience sample, you are unable to generalize the findings to all physicians in China.

Discussion…limitations…line 250…replace “didn’t” with “did not”

Author Response

Dear reviewer:

We are thankful for your recognition of our study, and feel appreciated with your honest comments and suggestions for improving our manuscript. Based on your comments and suggestions, we have made modification on our manuscript, re-written the discussion section, polished the language, and answered your questions below:

Methods

line 88: change “is” to “was” and “answer” to “answered” for correct tense. Also, please edit the end of the sentence: “…as end this study” it is hard to understand.

Re: Thanks for your suggestion. We have modified this sentence below.

L88: After sending electronic invitations to DXY member physicians (selected randomly by computer) via email and phone messages, a total of 1291 physicians responded and completed the online questionnaire. After excluding physicians who had never heard of e-cigarettes, the analysis included 1023 participants.

(2) lines 108-110 (and Table 2) since Likert items are ordinal measures, they cannot be statistically “averaged.” Please remove the average scores from your analysis and table. The findings from each item in the table are interesting enough, without needing to provide an average that is not possible statistically.

Re: Thanks for your suggestion. However, we could not find other indicators to represent the risk beliefs and attitude of physicians if remove the average scores from analysis. What’s more, we need to use the average score in multivariate logistic regression analyses. Therefore, we prefer to keep the average score, so that readers can have a clearer understanding of our results.

(3) I recommend deleting the analysis that includes never vs. ever smoking status. This doesn’t relate to your research study’s focus. Also, studies that ask smoking status of healthcare providers tend to look at current smoking status, not “ever” smoking. Ever smoking includes just trying a cigarette one time, which is much different that if the healthcare provider was a current smoker.

Re: Thanks for your suggestion. After discussion, we decide to classify smoking status into two categories: nonsmokers and former smokers vs current smokers, and reanalyze the data as following.

Characteristics

No. (%)

Smoking status

    Non-smoker and former smoker

Current smoker

896(87.6)

127(12.4)

Results

Table 2…please show the frequencies and percentages of agree, neutral, disagree for each Likert item.

Re: Thanks for your suggestion. Considering that in Chinese culture, if there is a neutral option, most people will choose this option, and we could not know their true attitudes toward e-cigarettes. Therefore, we don’t set neutral option of each Likert item, and participants can choose one of four answers ranging from strongly disagree to strongly agree.

Discussion

(1) lines 208-228…this entire paragraph should be deleted. This paragraph is based on your assumption that physicians think it’s important to discuss e-cigarettes with patients as a cessation tool. However, the survey item did not go into that level of detail. It simply asked about discussing e-cigarettes. The physicians could have interpreted the survey item as being important to discuss e-cigarettes because they are harmful.

Re: Thanks for your suggestion. We have modified the paragraph and just describe the finding in our survey.

More than two-thirds of physicians (61.3%) in our study thought it was important to discuss e-cigarettes with their patients, which indicated that a majority of Chinese physicians were aware of e-cigarettes and saw the issue of e-cigarettes as a priority. This result is in accordance with other studies of health care professionals. For example, Dong et al. reported that 67.6% of 185 lung cancer specialists thought it was important to communicate with patients about e-cigarettes [37]. In a survey of 561 Minnesota health providers who treat adolescents, Jessica et al. reported that 69% of respondents agreed that it was important to discuss e-cigarettes with adolescent patients[33]. Moreover, most of our participants (71.6%) were confident in their ability to answer their questions about e-cigarettes, with the prior study showing that nearly 50% of US primary care providers were confident in their knowledge about electronic cigarettes and ability to answer questions about them [38].

(2) discussion sections should discuss which findings from the study are worthy of discussion. So far, you have discussed that 8% of physicians used e-cigarettes, a paragraph based on an assumption that should be deleted, and a final paragraph that calls for more education for physicians about e-cigarettes. Please add a few paragraphs regarding the knowledge and attitudes that are concerning. These are important findings that should be discussed.

Re: Thanks for your suggestion. We delete the paragraph about e-cigarette use, and focus on discussing the beliefs, attitude, and confidence of Chinese physicians.

Despite high levels of confidence regarding answering patients’ questions on e-cigarettes, the results demonstrated a deficient knowledge of e-cigarettes’ health risks and a supportive attitude in the efficacy of e-cigarettes as smoking cessation aids among Chinese physicians. In our study, only 41.2% physicians agreed that e-cigarettes could induce other tobacco use, and 46.3% were aware of their adverse health effects, with 66.8% supporting the idea that e-cigarettes can be regarded as a type of smoking cessation treatment. Recent studies showed that most of the Korean lung cancer physicians (83.8%) believe that e-cigarettes could be a gateway to other tobacco use [37], and 62.3% of US physicians are concerned that e-cigarettes have adverse health effects [38]. In addition, other studies reported lower support for using e-cigarettes to quit smoking among physicians [37,39-41]. For example, only 18% of the specialties from Saint Louis University Hospital (USA) would advise e-cigarettes as cessation aid (i.e, as nicotine-replacement therapy) to patients [40]. Samuel et al. reported that more than a third (35%) of family physicians in Kansas were not sure about the effectiveness of e-cigarettes, and 12% thought e-cigarettes were either very effective or effective smoking cessation products [39]. Furthermore, authorities and major health organizations do not recommend the use of e-cigarettes as a regular smoking cessation tool. The US Preventive Service Task Force (USPSTF) noted that the evidence is insufficient to recommend e-cigarettes for quitting, and clinicians should direct patients to other cessation aids that are proven to be effective and safe [42]. This suggestion is shared by the American Heart Association [43], the American Academy of Family Physicians [44], and The Spanish Society for Pulmonary Medicine and Thoracic Surgery [45].

Limitations

(1) lines 248-249: the limitation is not about cause and effect, the limitation is that you had a convenience sample of physicians sampled through a website. Because it is a convenience sample, you are unable to generalize the findings to all physicians in China.

Re: Thanks for your suggestion. We have modified it below.

L248-249: First, the study only include physicians sampled through a website survey and so generalizing our findings to a broader base of Chinese physicians requires caution.

(2) line 250: replace “didn’t” with “did not”.

Re: Thanks for your suggestion, and we have replaced “didn’t” to “have not”.

Second, our analysis excluded physicians who have not heard of e-cigarettes, so the result may overestimate the e-cigarette usage rate, as well as the knowledge and confidence among Chinese physicians.

Reviewer 3 Report

MY COMMENTS ARE BELOW IN BOLD

Power computation

However, no study had shown that proportion among Chinese 81  physicians, and the previous study in the US showed that 20.5% of physicians communicated with 82  patients about e-cigarettes frequently in the clinic[31]. It is estimated that 660 physicians were needed 83  to obtain an α level of 5% with statistical power of 85%. Considering the possible variations, we 84  enlarged the sample size to 800 to be accurate. PLEASE DESCIBE WHAT CONSIDERING THE POSSIBLE VARIATIONS MEANS

Randomisation

After sending the electronic invitations to DXY member physicians randomly via email and 86 phone message, PLEASE DESCRIBE RANDOMISATION METHOD

Questionnaire

We developed a questionnaire based on previous research with similar topics in the 93 literature[32,33]. The questionnaire included questions regarding demographic information, 94  physicians’ knowledge toward e-cigarettes, attitudes toward e-cigarettes communication, attitudes 95  toward using e-cigarettes to quit, and confidence about e-cigarettes counseling. PLEASE DESCRIBE ANY RELIABILITY AND VALIDITY DATA ABOUT THE QUESTIONNAIRES FROM WHICH YOU ADAPTED THESE ITEMS

Adjusted analyses

3.3. Factors associated with the frequency of Chinese physicians asking their patients about e-cigarettes use 178  Multivariate analysis (Table 3) shows that high frequency of asking patients about e-cigarettes 179  use was significantly associated with physicians having received training about e-cigarettes 180 (OR=3.16), believing the fact that communicating with patients about e-cigarettes is important 181  (OR=1.42), and being confident about their ability to e-cigarettes counseling (OR=2.43). Additionally, 182  showing more positive attitude toward using e-cigarettes to quit (OR=0.79) significantly decreased 183  the odds of high frequency of asking patients about e-cigarettes use. PLEASE ALSO REPORT IMPORTANT NEGATIVE ASSOCIATIONS

Characteristics of respondents

There are at least two limitations in our study. First, the study is a cross-sectional survey and so 248 couldn’t make reliable conclusions regarding cause and effect. Second, our analysis excluded 249 physicians who didn’t hear of e-cigarettes, so the result may overestimate the e-cigarette using rate 250as well as the knowledge and confidence among Chinese physicians. PLEASE COMMENT TO WHAT EXTENT YOUR SAMPLE REPRESENTS THE PHYSICIAN POPULATION

Author Response

Dear reviewer:

We are thankful for your recognition of our study, and feel appreciated with your honest comments and suggestions for improving our manuscript. Based on your comments and suggestions, we have made modification on our manuscript and answered your questions below:

Power computation

However, no study had shown that proportion among Chinese physicians, and the previous study in the US showed that 20.5% of physicians communicated with patients about e-cigarettes frequently in the clinic [31]. It is estimated that 660 physicians were needed 83 to obtain an α level of 5% with statistical power of 85%. Considering the possible variations, we 84 enlarged the sample size to 800 to be accurate. PLEASE DESCRIBE WHAT CONSIDERING THE POSSIBLE VARIATIONS MEANS.

Re: Thanks for your question. We have checked and re-written how to calculate the sample size. We have to enlarge 660 to 1130, because 2015 China Adult Tobacco Survey showed that only 58.4% of physicians had heard of e-cigarettes.

P2 L86-87: The sample size was calculated by estimating the proportion of physicians who communicated with patients about e-cigarette use. However, no study has shown what proportion of Chinese physicians communicated with their patients. An earlier study in the US showed that 20.5% of physicians frequently communicated with patients about e-cigarettes in clinics [36]. It is estimated that 660 physicians are needed to obtain an α level of 5% with a statistical power of 85%. Considering that only 58.4% of physicians had heard of e-cigarettes in the 2015 China Adult Tobacco Survey [7], we had to enlarge the sample size to 1130 to be accurate.

Randomization

After sending the electronic invitations to DXY member physicians randomly via email and phone message, PLEASE DESCRIBE RANDOMIZATION METHOD

Re: Thanks for your question. We use computer code to select physicians randomly in the background of DXY website, and send electronic invitations to them.

After sending electronic invitations to DXY member physicians (selected randomly by computer) via email and phone messages, a total of 1291 physicians responded and completed the online questionnaire. After excluding physicians who had never heard of e-cigarettes, the analysis included 1023 participants. The study was approved by the Ethics Committee of the School of Public Health, Fudan University (IRB#2018-04-677). All participants gave electronic informed consent.

Questionnaire

We developed a questionnaire based on previous research with similar topics in the 93 literature [32,33]. The questionnaire included questions regarding demographic information, physicians’ knowledge toward e-cigarettes, attitudes toward e-cigarettes communication, attitudes toward using e-cigarettes to quit, and confidence about e-cigarettes counseling. PLEASE DESCRIBE ANY RELIABILITY AND VALIDITY DATA ABOUT THE QUESTIONNAIRES FROM WHICH YOU ADAPTED THESE ITEMS

Re: We are sorry that we didn’t show any data to confirm the reliability and validity. However, this questionnaire has been used in many studies before, such as, “Beliefs, Practices, and Self-efficacy of US Physicians Regarding Smoking Cessation and Electronic Cigarettes: A National Survey”, “Lung cancer specialist physicians' attitudes towards e-cigarettes: A nationwide survey”, and “Healthcare Providers' Beliefs and Attitudes About Electronic Cigarettes and Preventive Counseling for Adolescent Patients”. In addition, these studies didn’t describe any reliability and validity of their questionnaire. In strict sense, the questionnaire is not a scale, and we think these questions can reflect the beliefs, attitudes and confidence toward e-cigarettes among physicians according to previous studies.

Adjusted analyses

3.3. Factors associated with the frequency of Chinese physicians asking their patients about e-cigarettes use. Multivariate analysis (Table 3) shows that high frequency of asking patients about e-cigarettes use was significantly associated with physicians having received training about e-cigarettes (OR=3.16), believing the fact that communicating with patients about e-cigarettes is important (OR=1.42), and being confident about their ability to e-cigarettes counseling (OR=2.43). Additionally, showing more positive attitude toward using e-cigarettes to quit (OR=0.79) significantly decreased 183 the odds of high frequency of asking patients about e-cigarettes use. PLEASE ALSO REPORT IMPORTANT NEGATIVE ASSOCIATIONS

Re: Thanks for your suggestion. We decide to classify smoking status into two categories: nonsmokers and former smokers vs current smokers, and use SPSS to reanalyze the data. The results of our research are updated, and we have modified the paragraph according to your comments.

The multivariate analysis (Table 3) shows that a high frequency of asking patients about e-cigarette use is significantly associated with physicians having used e-cigarettes (OR=2.05), having received training about e-cigarettes (OR=3.13), and being confident about their ability to answer patients’ questions about e-cigarettes (OR=2.45). Additionally, showing a more positive attitude toward using e-cigarettes to quit (OR=0.79) significantly decreases the odds of high frequency of asking patients about e-cigarette use. However, gender, age, specialty, smoking status, risk beliefs of e-cigarettes are not associated with the frequency of Chinese physicians asking their patients about e-cigarette use.

Characteristics of respondents

There are at least two limitations in our study. First, the study is a cross-sectional survey and so 248 couldn’t make reliable conclusions regarding cause and effect. Second, our analysis excluded 249 physicians who didn’t hear of e-cigarettes, so the result may overestimate the e-cigarette using rate 250 as well as the knowledge and confidence among Chinese physicians. PLEASE COMMENT TO WHAT EXTENT YOUR SAMPLE REPRESENTS THE PHYSICIAN POPULATION.

Re: You really mentioned an important point. Firstly, the participants came from 31 provinces, and China only have 34 provinces. Secondly, the respondents’ smoking rate in our survey is 12.4%, which is close to the Chinese physicians’ smoking rate (14.6%) showed in 2015 China Adult Tobacco Survey. However, we collected our data through website and didn’t include physicians who have not heard of e-cigarettes. Therefore, generalizability to all Chinese healthcare providers will need to be established.